# A Survey of Lottery Ticket Hypothesis

## Abstract

The Lottery Ticket Hypothesis (LTH) states that a dense neural network model contains a highly sparse subnetwork (i.e., winning tickets) that can achieve similar performance as the original model when trained in isolation. While LTH has been proved both empirically and theoretically in many works, there still are some open issues, such as efficiency and scalability, to be addressed. Also, the lack of open-source frameworks and consensual experimental setting poses a challenge to future research on LTH. For the first time, we examine previous research and studies on LTH from different perspectives. We also discuss issues in existing works and list potential directions for further exploration. This provides an in-depth look at the state of LTH.

## 1 Introduction

Although deep neural network models have shown impressive performance in many domains such as computer vision and natural language processing, recent models such as GPT with billions of parameters are notoriously slow for training and inference (Brown et al., 2020). Pruning techniques can reduce the parameter counts of trained networks, decrease storage requirements, and improve computational performance of inference without compromising accuracy (LeCun et al., 1989; Liang et al., 2021b). Existing pruning technologies can be classified into two categories: structured pruning and unstructured pruning. Specifically, structured pruning (He et al., 2017; Liu et al., 2017; Li et al., 2017; Hu et al., 2016; Wen et al., 2016; Hong et al., 2018; Nonnenmacher et al., 2022; Halabi et al., 2022; Yin et al., 2023b; Nova et al., 2023) aims to find structural sparse patterns such as layer-wise, channel-wise, block-wise, and column-wise. Unstructured pruning (Han et al., 2015; Liu et al., 2019; Cao et al., 2019; Ren et al., 2019; Zhang et al., 2018; Alizadeh et al., 2022; Liao et al., 2023) removes individual weights instead of entire weight structures (e.g., channels). However, the sparse architectures produced by these pruning methods are difficult to train from the start, reaching lower accuracy than the original networks.

The Lottery Ticket Hypothesis (LTH) (Frankle & Carbin, 2018) states that a dense neural network model contains a highly sparse subnetwork (i.e., winning tickets) that can achieve even better performance than the original model. The winning tickets can be identified by training a network and pruning its parameters with the smallest magnitude in an iterative way or one-shot way. Before it is trained in each iteration, the weights will be reset to the original initialization. Instead of resetting to the original initialization, some works (Mikler, 2021; Renda et al., 2020a; Gadhikar & Burkholz, 2024) rewind the parameters of the subnetwork to the early stage of training. The identified winning tickets are retrainable to reduce the high memory cost and long inference time of the original neural networks, and the existence of winning tickets has been verified in both experiments and theory (Frankle et al., 2019; Diffenderfer & Kailkhura, 2021; Savarese et al., 2020; Zhou et al., 2019; Malach et al., 2020; Zhang et al., 2021a; Ma et al., 2021a; Zhou et al., 2019; Chen et al., 2021d; Su et al., 2020; Chen et al., 2022a). Later works have extended LTH to find the winning tickets for different kinds of neural networks such as generative models (Chen et al., 2021h; Kalibhat et al., 2021; Jiang et al., 2023), Transformers (Brix et al., 2020; Prasanna et al., 2020; Chen et al., 2020a; Behnke & Heafield, 2020), and GNNs (Chen et al., 2021c; Hui et al., 2023; Zhang et al., 2024; Harn et al., 2022) Despite the success of LTH, the iterative train-prune-retrain process and even the subsequent fine-tuning are costly to compute (Liu et al., 2021). Many works have proposed algorithms to find lottery tickets efficiently (Yin et al., 2023a; Zhang et al., 2021c; Wang et al., 2020b; Tanaka et al., 2020; Ye et al., 2020; Evci et al.,

Figure 1: Taxonomy of LTH

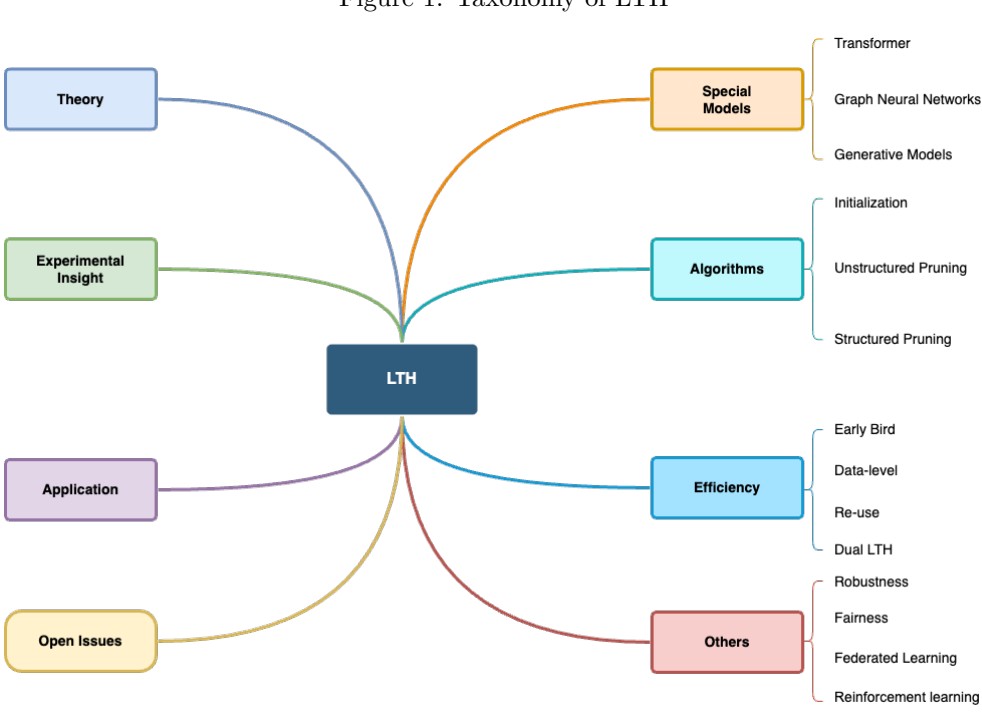

2020; Hou et al., 2022; Ma et al., 2022; Jaiswal et al., 2023a). LTH is also related to many other domains including robustness, transfer learning, fairness, and federated learning (Yuan et al., 2021; Goli & Aamodt, 2020; Raihan & Aamodt, 2020; Chen et al., 2021e; Mehta, 2019; Morcos et al., 2019b; Itahara et al., 2020; Seo et al., 2021; Mugunthan et al., 2022; Shi et al., 2023; Hansen & Søgaard, 2021). This survey aims to provide a comprehensive overview of the state of LTH research, categorizing various approaches, highlighting open issues, and presenting a benchmark.

In this paper, we first present a taxonomy of recent research that categorizes existing works into 8 topics: theory, special models, experimental insight, algorithms, efficiency, relations with other topics, open issues of existing works, and applications. Figure 1 shows the overview of the taxonomy. We first explain the theoretical foundations of the lottery tickets hypothesis. LTH has been studied on

Table 1: Terms

| Term | Description |
|------|-------------|
| Pruning | Removing neurons from a deep learning model |
| IMP | Iterative magnitude-based |
| Sparsity | The percentage of pruned parameters |
| Mask | Matrices to indicate the existence of neurons |
| Pruning ratio | The percentage of parameters to be pruned. |
| Initialization | The original value of model parameters |

three specific categories of models, i.e., Transformer, GNNs, and generative models. We show the difference regarding pruning between a general model and these specific models. While many novel algorithms have been developed, we discuss the effectiveness of different initialization methods and turning algorithms. In addition, we analyze the experimental insight in existing papers. Our survey also reviews efficient strategies that can reduce the high computational cost of iterative pruning. For each other topic (i.e., hardware, scalability, federated learning, and robustness), we show the connection with LTH. Lastly, we conclude our paper with open issues of LTH and potential research directions. We list terms with detailed descriptions in Table 1 to illustrate the terms widely used in the literature. All publications with available code are summarized in Table 2. We hope this survey serves as a valuable resource for LTH researchers and those seeking pruning technologies for various models.

## 2   Theory

We start with notations and functions related to LTH settings. Consider a network function $f(\cdot)$ that is initialized as $f(x, \theta_0)$ where x denotes input training samples and $\theta_0$ denotes the weights.

> **The lottery ticket hypothesis** – There exists an identically initialized subnetwork (i.e., winning tickets) that – when trained in isolation – can reach similar accuracy with the well-trained original network using the same or fewer iterations. The winning ticket can be identified by pruning its parameters with the smallest magnitude in an iterative way or one-shot way.

Based on the definition of LTH, we provide the following notations:

- Pretraining: train the network $f(x, \theta_0)$ for $T$ epochs, arriving at weights $\theta_T$.

- Pruning: generate a mask $m$ by pruning $s\%$ of weights in $\theta_T$.

- Re-initialization: reset the remaining weights to their values in $\theta_0$.

These three steps describe an interaction of magnitude- based pruning. According to the definition, the combination of $(\theta_0, m)$ is a winning ticket. The winning ticket can be found by pruning the weights with the smallest magnitude. The mechanism behind finding winning tickets is identified as three key inquiries (Paul et al., 2022): (1) Information Conveyed by IMP (iterative magnitude-based pruning) Mask: a mask $m$ obtained at the end of training carries crucial information about the identity of a specific subspace intersecting a desired linearly connected mode of a matching sublevel set; (2) SGD's Exploitation of Information: The stochastic gradient descent (SGD) optimization method demonstrates robustness that enables it to return to a desired mode despite significant perturbations early in training. (3) Role of Retraining in IMP: The retraining step in IMP is essential for finding a network with new, small weights to be pruned. These three keys explain why the winning ticket can be found.

Following the observation of LTH, an even more striking phenomenon (strong lottery tickets) is presented in (Ramanujan et al., 2020): not only do such sparse subnetworks exist at initialization but they already achieve impressive performance without any training. Specifically, this work asserts that proficient subnetworks can be derived from neural networks with randomly initialized weights, even without the necessity for weight training. Their proposed methodology entails the fixation of weights for each edge in the network, subsequently assigning a score to each edge. The scores are updated using the straight-through estimator and the top scores will be preserved. This approach does not require the utilization of pre-trained weight values, and the weights remain steadfast.

The existence of a winning ticket has been proved theoretically in many works. The first theoretical evidence of the existence of winning tickets was given by Malach et al. (2020). By approximating ReLU neurons, this work proves that optimizing the weights is equivalent to pruning entire neurons of a sufficiently large random network. Here ReLU referees to an activation with Rectified Linear Unit Agarap (2018). For every bounded distribution and every target network with bounded weights, a sufficiently over-parameterized neural network with random weights contains a subnetwork with roughly the same accuracy as the target network, without any further training. Given a target network to be pruned with $n$ neurons, polynomial overparameterization indicates that the relation between $n$ and the number of neurons in the pruned network is a polynomial function. However, the polynomial over-parameterization requirement in the proof is at odds with recent experimental results with networks that are wider than the bound. An exponential improvement to the over-parameterization requirement is offered in Pensia et al. (2020). Instead of providing significantly tighter bounds, (Orseau et al., 2020) points out that the overparameterized network only needs a logarithmic factor number of neurons per weight of the target subnetwork. For example, Given a target network with $n$ neurons, only $\log n$ neurons are necessary. Any network with a depth of $L$ and width of $W$ can be obtained by pruning a randomly initialized network with a depth of $2L$ and width increased by a logarithmic factor, or a network with a depth of $L + 1$ and width increased by another logarithmic factor (Burkholz, 2022b). This discovery implies that effective lottery tickets can be identified at frequently used depths, necessitating only logarithmic levels of overparameterization.

Table 2: LTH-related publications with code

| Method | Conference | Framework | Code | Model | Datasets |
|---|---|---|---|---|---|
| LTH (Frankle & Carbin, 2018) | ICLR | Tensorflow | https://github.com/google-research/lottery-ticket-hypothesis | VGG-19, ResNet-18 | MNIST CIFAR10 |
| Robust Tickets (Zheng et al., 2022) | ACL | Pytorch | https://github.com/ruizheng20/robust_ticket | BERT | IMDB AGNEWS SST-2 |
| Super Tickets (Liang et al., 2021a) | ACL | Pytorch | https://github.com/RICE-EIC/SuperTickets | BERT | GLUE |
| Early BERT (Chen et al., 2020c) | ACL | Pytorch | https://github.com/VITA-Group/EarlyBERT | BERT | MNLI QNLI QQP SST-2 |
| Eearly-GCN (You et al., 2022b) | AAAI | Pytorch | https://github.com/RICE-EIC/Early-Bird-GCN | GCN | Cora Citeseer Pubmed |
| Edge-popup (Ramanujan et al., 2020) | CVPR | Pytorch | https://github.com/allenai/hidden-networks | VGG ResNet | Cifar-10 Cifar-100 Tiny-ImageNet |
| Life-long ticket s(Chen et al., 2020b) | ICLR | Pytorch | https://github.com/VITA-Group/Lifelong-Learning-LTH | ResNet18 | Cifar-10 ImageNet |
| Elastic LTH (Chen et al., 2021f) | NeurIPS | Pytorch | https://github.com/VITA-Group/ElasticLTH | ResNet-20 ResNet-18 | Cifar-10 Cifar-100 |
| DLRA (Schotthöfer et al., 2022) | NeurIPS | Tensorflow Pytorch | https://github.com/ScSteffen/DLRT-Net https://github.com/COMPiLELab/DLRT-Net | LeNet5, ResNet-50 AlexNet, VGG16 | MNIST Cifar-10 ImageNet1K |
| CS (Savarese et al., 2020) | NeurIPS | Pytorch | https://github.com/lolemacs/continuous-sparsification | VGG-16 ResNet-20 | Cifar-10 |
| DLTH (Bai et al., 2022) | ICLR | Pytorch | https://github.com/yueb17/DLTH | ResNet56 ResNet18 | Cifar-10 Cifar-100 ImageNet |
| Early Bird (You et al., 2020) | ICLR | Pytorch | https://github.com/GATECH-EIC/Early-Bird-Tickets | PreResNet101 VGG16 | Cifar-10 Cifar-100 |
| Rewinding (Renda et al., 2020b) | ICLR | Tensorflow | https://github.com/lottery-ticket/rewinding-iclr20-public | ResNet-56 ResNet-34 ResNet-50 GNMT | Cifar-10 ImageNet WMT6 EN-DE |
| ULTH (Burkholz et al., 2022) | ICLR | Pytorch | https://github.com/RelationalML/UniversalLT | Conv2 | Cifar-10 Imagenet |
| SuperTickets (You et al., 2022a) | ECCV | Pytorch | https://github.com/GATECH-EIC/SuperTickets | Transformer | ImageNet Cityscapes ADE20K |
| ULTH-GNN (Chen et al., 2021c) | ICML | Pytorch | https://github.com/VITA-Group/Unified-LTH-GNN | GNN | Cora Citeseer Pubmed |
| PrAC (Zhang et al., 2021c) | ICML | Pytorch | https://github.com/VITA-Group/PrAC-LTH | ResNet VGG | Cifar-10 Cifar-100 Tiny-ImageNet |
| RigL (Evci et al., 2020) | ICML | Tensorflow Pytorch | https://github.com/google-research/rigl https://github.com/nollied/rigl-torch | MobileNet-v1 ResNet-50 | ImageNet2021 Cifar-10 Mnist |
| Sanity Checks (Ma et al., 2021a) | NeurIPS | Pytorch | https://github.com/boone891214/sanity-check-LTH | VGG ResNet | CIFAR-10 CIFAR-100 Tiny-ImageNet ImageNet-1K |
| Object Recognition (Girish et al., 2021) | CVPR | Pytorch | https://github.com/boone891214/sanity-check-LTH | RCNN ResNet | COCO |
| LTH-BERT (Prasanna et al., 2020) | EMNLP | Pytorch | https://github.com/sai-prasanna/bert-experiments | BERT | GLUE |
| SubsetSum (Pensia et al., 2020) | NeurIPS | Pytorch | https://github.com/acnagle/optimal-lottery-tickets | LeNet5 | MNIST |

The proof is also extended to convolutional neural networks (CNN) by (da Cunha et al., 2022): with high probability, it is possible to approximate any CNN by pruning a random CNN whose size is larger by a logarithmic factor. Considering that most theoretical validations regarding the existence of winning tickets have focused on networks with the ReLU activation functions, (Burkholz, 2022a) demonstrate that winning tickets are likely to be present in modern architectures featuring convolutional and residual layers, which incorporate a wide range of activation functions. A unifying framework is also introduced to prove the strong lottery ticket hypothesis for MLPs, CNNs, and general equivariant networks in (Ferbach et al., 2022). The strong lottery ticket hypothesis is extended functions that maintain the operation of the group, known as G-equivariant networks. The study establishes, with a high likelihood, that it is possible to approximate any G-equivariant network with fixed width and depth by pruning a randomly initialized over-parametrized G-equivariant network to obtain a G-equivariant subnetwork. Additionally, the research demonstrates that the proposed over-parametrization approach is optimal and sets a lower limit on the number of effective parameters relative to the error tolerance.

In an alternative way, Zhang et al. (2021b) apply the dynamical systems theory and inertial manifold theory to theoretically validate the effectiveness of the Lottery Ticket Hypothesis. Moreover, they demonstrate lossless pruning and one-shot pruning in theory. Gradient dynamical systems are employed to elucidate optimization problems in neural networks, and project high-dimensional systems onto inertial manifolds, yielding a low-dimensional system specific to pruned subnetworks. PAC-Bayesian theory has also been

leveraged to elucidate the relationship between the Lottery Ticket Hypothesis (LTH) and its generalization ability (Sakamoto & Sato, 2022). It turns out that using an initial large learning rate during pruning leads to suboptimal performance of the final tickets. Also, the authors observe that an initial large learning rate causes the optimizer to converge to flatter minima. It is hypothesized that winning tickets are associated with relatively sharp minima, which is typically considered detrimental to generalization. Employing a spike-and-slab distribution in the PAC-Bayes bound analysis of winning tickets, the authors discover that fatness contributes to improved accuracy and robustness. Furthermore, the distance to the initial weights significantly impacts the winning tickets. A sparse deep neural network can be represented with a sparse graph and the connectivity in the pruned model can be characterized by the spectral expansion property in the graph. The validity of LTH has been studied based on Ramanujan graph properties in terms of certain spectral bounds (Pal et al., 2022). Accordingly, a modification of existing iterative pruning algorithms that preserves the Ramanujan graph property is proposed for more efficient winning ticket search.

## 3 Special Models

It is non-trivial to extend the lottery tickets hypothesis to special models such as graph neural networks, transformers, and generative models since these models have distinctive architectures. In this section, we will elaborate on the variants of LTH for these models.

### 3.1 Graph Neural networks

A distinctive characteristic of Graph Neural Networks (GNNs) Kipf & Welling (2017); Hamilton et al. (2017); Hui et al. (2020) is that a graph structure is fed into the model. Intuitively, the computational resource consumption in both training and inference of GNNs will increase substantially as the size and complexity of graphs grow. Also, redundant or noisy edges may exist in the graph structure. To tackle this issue, UGS (Chen et al., 2021c) introduced a unified GNN sparsification (UGS) framework. This framework concurrently prunes the graph adjacency matrix and model weights, thereby expediting GNN inference on large-scale graphs. Expanding on the UGS framework, they extended the lottery ticket hypothesis to GNNs, defining a graph lottery ticket (GLT) as a combination of a pruned graph and a sparse sub-network. GLTs can be discerned from the original GNNs and the densely connected graph through the iterative sparsification process based on mask weights.

However, GLT requires a full pretraining process, which may not be universally applicable in real-world scenarios. Additionally, GLT's sparsification approach may lead to significant information loss. To address these limitations, (Wang et al., 2022b) propose the Dual Graph Lottery Ticket (DGLT) framework. This approach transforms a random ticket into a graph lottery ticket, allowing for a more comprehensive exploration of the relationships between the original network/graph and the sparse counterpart. Through regularization-based network pruning and hierarchical graph sparsification, DGLT achieves a triple-win scenario: high sparsity, commendable performance, and enhanced explainability. (Hui et al., 2023) also propose a novel approach to address the limitations of UGS. They observe that the performance of a sparsified GNN significantly declined as graph sparsity increased beyond a certain threshold. To mitigate this, they introduce two enhancements: First, they refine the adjacency matrix pruning by incorporating an auxiliary loss head to guide edge pruning more comprehensively. Second, they treat unfavorable graph sparsification as adversarial data perturbations, formulating the pruning process as a min-max optimization problem to enhance robustness. Instead of pruning based on magnitude, an algorithm is proposed in Harn et al. (2022) to prune weights and edges based on gradient. Similar to GLT, GEBT (Graph early-bird tickets) (You et al., 2022b) develops a training framework to boost the efficiency of GNNs training by enabling co-sparsification.

Previous GLT methods use the same topology structure across all layers. The strategy hinders the integration of GLT into deeper and larger-scale GNN contexts. To address this issue, an Adaptive, Dynamic, and Automated framework (Zhang et al., 2024) is introduced for identifying Graph Lottery Tickets (AdaGLT). The proposed method tailors layer-adaptive sparse structures for various datasets and GNNs and captures GLT across diverse sparsity levels. The theoretical proofs to mitigate over-smoothing issues and obtain improved sparse structures in deep GNNs are also provided.

### 3.2 Transformer

Large pre-trained transformers have achieved great success in the past few years. However, due to the exponentially increasing parameter counts of these large pre-trained models, fine-tuning these models with non-industry standard hardware is becoming seemingly impossible. (Yu et al., 2019) conduct a comprehensive investigation into the applicability of the Lottery Ticket Hypothesis (LTH) within the domains of Natural Language Processing (NLP). They scrutinize recurrent LSTM models as well as large-scale Transformer models. Notably, the authors successfully identify initializations that qualified as winning tickets, facilitating Transformer models to achieve performance levels nearly equivalent to their full-size counterparts, despite a reduction in size by one-third. Two studies have addressed the feasibility of applying the LTH to BERT. (Prasanna et al., 2020) successfully identified subnetworks within fine-tuned BERT that exhibited performance on par with the complete model. They observed that similarly sized subnetworks sampled from other parts of the model demonstrated worse performance. However, they noted that even the least performing subnetworks exhibited notable trainability, indicating the substantial utility of the majority of pre-trained BERT weights. In a separate investigation, (Chen et al., 2020a) examined pre-trained BERT and unearthed the presence of trainable and transferable subnetworks within it. Notably, these subnetworks were discovered during the pre-training phase, distinguishing the earlier research, which obtained subnetworks during the fine-tuning phase, a process necessitating a certain degree of additional training. In addition to its validation in the field of NLP, (Gan et al., 2022) explore whether trainable subnetworks, as suggested by the LTH, exist in vision-and-language models. The findings indicate that while exact matches are challenging, "relaxed" winning tickets at 50%-70% sparsity maintain 99% accuracy. Task-specific pruning leads to reasonably transferable subnetworks, and pre-training tasks at 60%/70% sparsity achieve 98%/96% accuracy. (Behnke & Heafield, 2020) investigate the pruning of attention heads in Transformers, prompted by recent research indicating a lack of confidence in the decisions made by a significant portion of attention heads. Through experiments in machine translation, they demonstrate the potential to remove up to three-quarters of attention heads from transformer-big during early training.

During the fine-tuning stage of Transformer models applied to NLP tasks, the integration of adapter modules has been found to expedite transfer learning. However, adapter fusion models, which incorporate knowledge from various tasks, can lead to redundancy and increased computational costs. To mitigate this, (Wu et al., 2022) propose a method to assess the impact of each adapter module and employ the Lottery Ticket Hypothesis to prune them effectively. (Gong et al., 2022) explore the applicability of the Lottery Ticket Hypothesis to pre-trained language models (PLMs) for fine-tuning. The authors introduce a regularization approach using L1 distance and identify a subnetwork structure termed the "dominant winning ticket." They empirically demonstrate that this dominant winning ticket can achieve performance comparable to the full-parameter model, is transferable across different tasks, and exhibits a natural structure within each parameter matrix. To address the computational cost of repetitive training and pruning routine of the transformer, Instant Soup Pruning (ISP) (Jaiswal et al., 2023b) is introduced as an efficient alternative to the iterative magnitude pruning (IMP) method. ISP leverages the concept of model soups, merging fine-tuned weights from multiple models to achieve better minima. The proposed approach replaces the costly intermediate stages of IMP with a computationally efficient weak mask generation and aggregation process. Specifically, ISP generates multiple weak and noisy subnetworks through a few iterations with varying training protocols and data subsets, then combines them to create a high-quality denoised subnetwork.

### 3.3 Generative Models

While prior research mainly focused on supervised learning scenarios, (Kalibhat et al., 2021) extend the concept to deep generative models including GANs and VAEs. The study demonstrates that iterative magnitude pruning, combined with generative losses, effectively identifies winning tickets, achieving high sparsity levels. Moreover, the transferability of winning tickets across different generative models implies they have inherent biases beneficial for training various deep generative models. Furthermore, the paper introduces the concept of "early-bird tickets," enabling substantial reductions in floating-point operations and training time, enhancing the feasibility of training large-scale generative models under resource constraints.

(Chen et al., 2021h) validate the presence of trainable matching subnetworks in various deep GAN architectures. Additionally, the results suggest that pruning the discriminator has a minor impact on the existence and quality of matching subnetworks, while the initialization weights in the discriminator significantly influence the outcome. Furthermore, the study demonstrates the robust transferability of these subnetworks to unseen tasks. Drawing inspiration from previous findings on independently trainable and highly sparse subnetworks within GANs, (Chen et al., 2021a) introduce a novel approach to data-efficient GAN training. The proposed method involves identifying a lottery ticket from the original GAN using a small training set of real images, and then focusing on training that sparse subnetwork using the same set. The coordinated framework demonstrates significant improvements over existing real image data augmentation methods and introduces a novel feature-level augmentation that can be used in conjunction with them.

An extensive exploration of the applicability of pruning techniques in generative models is conducted by Yeo et al. (2023) to discover subnetworks that perform similarly to or better than the trained dense model. They introduce the concepts of weak lottery tickets and strong lottery tickets. The former refers to tickets identified from networks with learned weights, while the latter denotes tickets identified without any weight updates. The experimental result shows that a subnetwork of GFMN with only 10% of the weights remaining can reach similar FID scores as GFMN. LTH has also been applied to diffusion models (Jiang et al., 2023). It shows that we can find a winning ticket of DDPM (Ho et al., 2020) at sparsity $90\% - 99\%$ without compromising performance for denoising diffusion probabilistic models on benchmarks (Cifar-10, Cifar-100, MNIST). Different from existing LTH works which identify the subnetworks with a unified sparsity along different layers, this work proposes to find the winning ticket with varying sparsity along different layers in the model.

## 4 Experimental Insight

The discrepancies regarding LTH have emerged due to different settings and several works have investigated the criteria to identify winning tickets. In this survey, we conclude several experimental insights.

### 4.1 How much can be pruned?

Although the existence of winning tickets has been verified in theory, how much weight can be pruned is still an open question in an experimental setting. The question is answered in (Ye et al., 2020) to find a subnetwork while tolerating an acceptable level of accuracy reduction. They propose a pruning method based on greedy optimization. This method results in an exponential decrease in the discrepancy between the pruned network and the original network as the number of winning tickets decreases. (Liang et al., 2021a) identify a phase transition phenomenon, where the performance of winning tickets initially improves with increased compression, but then starts to deteriorate after a threshold. These threshold tickets are referred to as "super tickets." The presence and characteristics of this phase transition are influenced by both the task and the model size, with larger models and smaller datasets intensifying the effect. Experimentation on the GLUE benchmark demonstrates that super tickets enhance single-task fine-tuning, and adaptively sharing them across tasks benefits multi-task learning.

As an increasing number of researchers engage in the investigation of the LTH, discrepancies have emerged in the experimental design and criteria formulation for the identification of winning tickets. To resolve these issues, (Ma et al., 2021b) proposes a more rigorous definition of the lottery ticket hypothesis and provides concrete evidence of its applicability across various DNN architectures and applications. The study quantitatively explores the relationships between winning tickets and various experimental factors. The authors discover that crucial training hyperparameters, such as learning rate and training epochs, along with architectural features like capacities and residual connections, significantly influence the identification of winning tickets. Based on their findings, the study offers a guideline for parameter settings tailored to specific architecture characteristics.

## 4.2 Pruning more in deep layers

In an investigation of applying LTH to computer vision models (Girish et al., 2021), two pruning methods are introduced: layer-wise pruning and global pruning. It has been observed that when sparsity levels are low, there is a marginal disparity in accuracy between the two pruning methods. However, as sparsity levels increase, global pruning exhibits a slight performance advantage. This is attributed to the uneven sparsity distribution across layers in the case of global pruning, leading to enhanced performance of subnetworks. Through an analysis of sparsity rates in each layer, they also observe that deeper layers are pruned more. Similar results are also observed in (Jiang et al., 2023) where an incremental rate is used to increase the pruning ratio in deeper layers.

## 4.3 Key factors

There are three critical components in LTH: Zeros, Signs, and the Supermask (Zhou et al., 2019). Three questions are answered by ablating these factors: why setting weights to zero is important, how signs are all you need to make the reinitialized network train, and why masking behaves like training. It is the sign in the original initialization, not the magnitude of the weights, that is crucial to the performance of LT networks. Also, the masking procedure can be considered as a training operation. Supermasks are observed to produce partially working networks without training. (Evci et al., 2022) emphasize that training unstructured sparse Neural Networks (NNs) from random initialization often results in poor generalization, with Lottery Tickets and Dynamic Sparse Training being notable exceptions. They observe that in the initialization phase of sparse NNs, gradient flow tends to be weak. In comparison to other methods, Dynamic Sparse Training significantly bolsters gradient flow during the training phase, potentially contributing to its effectiveness. Conversely, Lottery Tickets does not exhibit a significant improvement in gradient flow. Therefore, the authors posit three key factors contributing to the success of Lottery Tickets: 1. Proximity to the pruning solution during initialization. 2. Residing within the basin of the pruning solution. 3. Learning functions akin to the pruning solution.

# 5 Algorithms

There are two core parts in the pruning algorithm: initialization and pruning. Following the original IMP-based pruning, many works have proposed new methods regarding both initialization and pruning.

## 5.1 Initialization

Typically, pruning involves training, compressing, and then retraining to recover accuracy. The conventional method of fine-tuning uses a low fixed learning rate to train the unpruned weights from their final trained values. In contrast, (Renda et al., 2020b) introduce two alternative approaches: weight rewinding, which reverts unpruned weights to earlier training stages, and learning rate rewinding, which applies the original learning rate schedule to the unpruned weights. Both rewinding techniques surpass fine-tuning, forming the foundation of a network-agnostic pruning algorithm that achieves comparable accuracy and compression ratios to specialized state-of-the-art techniques. It is important to understand how learning rate rewinding (LRR) excels in pruning schemes and parameter learning because it can bring us closer to the design of more flexible deep learning algorithms. To this end, Gadhikar & Burkholz (2024) conducts experiments to investigate the ability of LRR to flip parameter signs early and stay robust to sign perturbations.

## 5.2 Unstructured pruning

Most of the existing works in the literature focus on finding unstructured winning tickets where the positions of zeros are not predefined. These methods pruned individual elements with the smallest magnitude. (Shang et al., 2022) delve into the Magnitude-Based Pruning (MBP) scheme and provide a fresh perspective by employing Fourier analysis on the deep learning model to guide model selection. In addition to elucidating the generalization capability of MBP using the Fourier transform, a novel two-stage pruning approach is

proposed. The first stage involves obtaining the topological structure of the pruned network, followed by a second stage of retraining to restore capacity using knowledge distillation from lower to higher frequencies.

Different from these methods based on IMP, (You et al., 2022a) present an approach addressing Deep Neural Networks (DNNs) termed "Two-in-One Training," which enables the direct identification of efficient DNNs and their associated lottery subnetworks, referred to as SuperTickets, from a supernet. Previous methods typically employ Neural Architecture Search (NAS) followed by pruning algorithms to obtain lottery subnetworks for efficient DNNs. This approach, known as "First-search-then-prune (S+P)," involves an initial search for an efficient network architecture and subsequent sparsity induction. In contrast, this paper proposes a unified approach that simultaneously conducts architecture search and pruning during the training of the supernet within NAS. In the search phase, the authors compute importance factors for search units and set a threshold $\epsilon$ to remove unnecessary units. In the pruning phase, they employ the Importance Masking Pruning (IMP) method to achieve the desired level of sparsity. Additionally, two techniques are introduced in the search and pruning phases. In the search phase, Iterative Reactivation is implemented. Unlike the conventional IMP method, the authors incorporate reactivation during the search process rather than after pruning. This choice is made to retain training opportunities for the sparsified network, which is believed to contribute to performance enhancement. Experimental results validate this hypothesis by comparing the performance of networks with reactivation during search versus pruning. Schotthöfer et al. (2022) introduce a novel approach to identify efficient low-rank subnetworks. These subnetworks are determined and adjusted during the training process, significantly reducing the time and memory resources required for both training and evaluation. The key concept involves restricting weight matrices to a low-rank manifold and updating the low-rank factors instead of the entire matrix during training. This method ensures that the training updates stay within the specified manifold, providing guarantees of approximation, stability, and descent.

Continuous Sparsification (CS) is introduced as another alternative to the traditional IMP approach. Unlike IMP, CS does not require weight resetting between epochs, making it a more efficient method for subnetwork discovery. Additionally, CS allows for dynamic adjustment of sparsity during the training process, thus achieving a better balance between model accuracy and sparsity. In contrast to traditional methods, CS introduces $L_1$ regularization to encourage most elements in the weight vector to be zero:

$$\min_{\theta \in \mathbb{R}^d, m \in \{0,1\}^d} L(f(\cdot; m \odot \theta)) + \lambda \cdot \|m\|_1 \tag{1}$$

Similar to most methods that utilized regularization, to avoid the discrete space of $m$, they re-parameterize $m$ by introducing a new variable $s \in \mathbb{R}^d$. To mitigate biased and/or noisy training caused by gradient estimators, they define the Heaviside step function such that $m := H(s)$, with $s \in \mathbb{R}^d_{\neq 0}$ and $H : \mathbb{R}^d_{\neq 0} \to \{0,1\}$, i.e., $H(s) = 1$ if $s > 0$ and 0 otherwise. This leads to an equivalent form to solve equation (1):

$$\min_{w \in \mathbb{R}^d, s \in \mathbb{R}^d_{\neq 0}} L(f(\cdot; H(s) \odot \theta)) + \lambda \cdot \|H(s)\|_1 \tag{2}$$

Due to the discontinuity of the step function $H$ at certain points, and its derivative being zero at all points, dealing with $H$ in computer programs poses some challenges. The authors propose an approximation method, using an S-shaped function parameterized by $\beta$ to approximate $H$. The form of this approximating function is $s \mapsto \sigma(\beta s)$, where $\sigma$ is the sigmoid function. Thus, the final objective function is:

$$L_\beta(\theta, s) := L(f(\cdot; \sigma(\beta s) \odot \theta)) + \lambda \cdot \|\sigma(\beta s)\|_1 \tag{3}$$

During training, gradient descent is used to minimize the loss function $L_\beta(\theta, s)$ to learn the sparse network. Simultaneously, $\beta$ is jointly adjusted in the process of $T$ parameter updates. After each update, corresponding parameter values $\theta^{(T)}$, $s^{(T)}$, and $\beta^{(T)}$ are obtained. Thus, between each iteration, only $s$ is adjusted based on the changes in parameter values, without modifying the network weights. This leads to higher efficiency in finding networks, and the continuous dynamic adjustments ensure the smoothness of sparsity and the accuracy of the network.

## 5.3 Structured pruning

Although LTH can find element-wise sparse subnetworks and even outperform the original dense models, unstructured pruning is not practical for real hardware acceleration. "Generalized Lottery Ticket Hypothesis" is investigated by Alabdulmohsin et al. (2021) to encompass both unstructured and structured pruning under a common framework. Specifically, structured pruning is achieved by carefully selecting the basis and using algorithms originally developed for unstructured pruning such as IMP. To address the limit of winning tickets in practice, (Chen et al., 2022a) propose a "post-processing technique" after each round of unstructured pruning to enforce the structural sparsity. Specifically, the pruned elements are re-filled back in some channels, and then non-zero elements are re-grouped to create flexible group-wise structural patterns. It turns out that the hardware acceleration roadblock of LTH can be removed. (Li et al., 2023) attempt to alleviate the constraints on network pruning by employing a mixed-precision quantization approach. They validate the existence of quantized lottery tickets, which are denoted as MPQ-tickets (Mixed-Precision Quantization). Additionally, the study demonstrates that MPQ lottery models exhibit greater flexibility compared to regular lottery models, and they derive more substantial benefits from pruning when compared to quantized neural networks (QNNs).

# 6 Efficiency

Despite the success of LTH, the iterative train-prune-retrain process is time-consuming. To address this issue, many works propose to find the winning tickets in the early state. Some other works reduce the computation cost by reducing the size of data or using efficient search algorithms. Remarkably, the possibility of re-using an identified winning ticket across different datasets or architectures has been investigated.

## 6.1 Early Bird Tickets

The concept of Early-bird (EB) tickets is first introduced by You et al. (2020). It refers to winning tickets identified in the early stages of training using cost-effective approaches like early stopping and low-precision training at high learning rates. This discovery aligns with prior observations that critical network patterns emerge early. Additionally, they introduce a mask distance metric for efficiently identifying EB tickets with minimal computational overhead, without requiring knowledge of the final winning tickets. By capitalizing on the existence of EB tickets and employing the proposed mask distance, we develop efficient training methods that involve identifying EB tickets through low-cost techniques and then continuing training exclusively on these EB tickets to reach the target accuracy. Chen et al. (2020c) introduce EarlyBERT, drawing inspiration from the concept of Early-Bird Lottery Tickets proposed by (You et al., 2022b). The study put forth a computationally efficient training algorithm applicable to both pre-training and fine-tuning large-scale language models. By slimming down specific components of a transformer, the study identifies structured winning tickets in the early stages of BERT training, leading to efficient BERT training.

In the application of the LTH to object recognition tasks, (Girish et al., 2021) also utilizes Early-Bird tickets. They observe that after the middle stage, the mask stabilizes. This observation suggests that a substantial reduction in the number of training iterations in computer vision tasks can be achieved with minimal impact on network performance. (You et al., 2022b) is the first to identify early birds in Graph Convolutional Networks (GCN) and introduce a novel concept known as Graph Early-Bird (GEB) tickets. These emerge in the early stages of sparsifying GCN graphs. They also propose an effective detector to automatically identify the presence of GEB tickets. Additionally, they advocate for graph-model co-optimization and present a highly efficient training framework called GEBT. This framework enhances the efficiency of GCN training by jointly identifying early-bird tickets in both GCN graphs and models while enabling simultaneous sparsification of both components. (Shen et al., 2022) present an approach similar to early-bird tickets. They introduce an Early Pruning Indicator (EPI), which distinguishes itself from early-bird tickets by utilizing sub-network architectural similarity to initiate pruning once the architecture reaches stability.

Expanding on the Early-Bird (EB) approach, (Kim et al., 2022) introduces a novel method termed Early-Time (ET). They establish the stopping criterion for training by evaluating the Kullback–Leibler (KL) divergence between class prediction distributions at distinct time intervals. Once the KL divergence reaches a

predetermined threshold, it signifies the acquisition of winning tickets. The ET method can be synergistically employed with EB, resulting in a reduction of training time by up to 38

Instead of pruning the trained networks, (Wang et al., 2020a) endeavor to prune networks during the initialization phase to save computational costs during training. They introduce a novel criterion known as Gradient Signal Preservation (GraSP). This method involves the removal of weights that have the least impact on reducing the gradient norm after pruning. (Rachwan et al., 2022) further refine the Gradient Signal Preservation (GraSP) by introducing Early Compression via Gradient Flow Preservation (EarlyCroP). This approach prioritizes the preservation of Gradient Flow (GF) and leverages the close relationship between the Neural Tangent Kernel (NTK) and GF. Consequently, it allows for network pruning with minimal disruption to the NTK, ensuring high levels of sparsity while maintaining performance comparable to that of the dense network. EarlyCroP requires only a single model training session and can be implemented either before or early in the training process.

## 6.2   Reduce Data Size

An alternative way to find lottery tickets efficiently is using a selected subset of data rather than using the full training set. With the small dataset, the cost of interactive pruning can be reduced significantly. (Zhang et al., 2021d) propose reducing the size of the training set to more efficiently acquire winning tickets. They specifically select a subset of data, termed the Pruning-Aware Critical set (PrAC set), instead of employing the entire training dataset. The PrAC set encompasses samples that are considered to be the most challenging and information-rich for dense models. Training and pruning networks using the PrAC set enables the discovery of high-quality winning tickets, while concurrently reducing the number of iterations needed for training. (Shen & Xing, 2022) introduce the "dataset lottery ticket hypothesis," suggesting the presence of a subset within an extensive large-scale dataset. On this subset, the performance disparities (e.g., the difference in terms the accuracy) under various training frameworks, hyperparameters, and other conditions are especially significant. Consequently, it facilitates a substantial reduction in resource demands and enhances the computational efficiency of experiments, all while maintaining accuracy on the complete dataset. They employ multiple approaches, including a uniform selection scheme commonly utilized in the literature, as well as the creation of subsets based on prior knowledge, to validate this hypothesis. (Shen et al., 2023) face difficulties in locating subnetworks within Vision Transformers (ViTs) at the weight level using conventional methods. Consequently, they attempt to sparsify the input data, particularly image patches. In essence, there exists a specific subset of input image patches. When a ViT is exclusively trained using this subset, it attains a level of accuracy similar to ViTs trained with the entire set of patches. They term this subset of input patches as "em winning tickets", signifying the genuine information that is meaningful to the tasks within the input data (i.e., an important subset related to the task).

## 6.3   Re-use winning tickets

Since the "retrainability" of a winning ticket is the most distinctive property of LTH, an intuitive method to reduce the cost is re-using the winning tickets across various tasks and models. (Morcos et al., 2019a) explore the potential of reusing winning tickets across different datasets and optimizers. They observe that winning tickets generated from larger datasets tend to transfer better. Moreover, these initializations also show promise across different optimization methods. These results indicate that winning ticket initializations from larger datasets possess inductive biases that benefit neural networks more broadly, offering promise for improved initialization methods. (Van Soelen & Sheppard, 2019) lay the foundation for a transfer learning technique that distills the original network to its crucial connections, obviating the need to freeze entire layers. The authors show that in the case of CNNs, networks can not only be pruned down to a mere 10% of their original parameters but these sparsified networks can also be retrained on similar datasets with only a minor decrease in accuracy. Rooted in the LTH, this approach presents a promising alternative to conventional transfer learning methods and has shown encouraging initial outcomes. Burkholz et al. (2022) also theoretically prove the existence of universal lottery tickets. These universal lottery tickets can be directly reused across various tasks without the need to find a new lottery ticket for the new task again. (Desai et al., 2019) investigate the feasibility of retraining a sparse subnetwork obtained from one domain to a dissimilar domain and explores the impact of different initialization strategies during transfer. The

experiments reveal that subnetworks obtained through lottery ticket training possess an inductive bias that transcends specific domains, enabling effective application across diverse domains.

However, not all subnetworks exhibit universal transferability. (Chen et al., 2020a) investigate subnetworks extracted in the context of masked language modeling tasks, which demonstrate broad transferability. In contrast, subnetworks identified in other NLP tasks may possess limited applicability or even lack transferability altogether when applied in a transfer learning context. In addition to specific studies on the knowledge transfer of LTH in particular tasks and domains, there are ongoing investigations into the model's capacity for continuous learning, known as lifelong learning. (Chen et al., 2020b) extend the LTH into the realm of lifelong learning, introducing the concept of "lifelong tickets." This research showcases that lifelong tickets can enhance learning performance across continual tasks. However, pruning networks in the lifelong setting presents significant challenges. Firstly, conducting greedy weight pruning in sequential tasks is hindered by a bias towards earlier tasks. Secondly, the compact network capacity of tickets in lifelong learning exacerbates the issue of catastrophic forgetting. To address these, two pruning options are proposed: top-down and bottom-up. The top-down approach extends iterative pruning over sequential tasks. In contrast, the bottom-up approach, allowing dynamic adjustment of model capacity, proves effective in mitigating early-stage excessive pruning. Additionally, the study introduces "lottery teaching," a method leveraging knowledge distillation with external unlabeled data to further counteract forgetting.

(Sun et al., 2020) present a novel parameter sharing mechanism, "Sparse Sharing," for deep multi-task learning. Unlike conventional approaches relying on predefined sharing schemes, Sparse Sharing automatically identifies a sparse sharing structure tailored to the specific task relations. It starts with an over-parameterized base network, from which each task extracts a subnetwork. These subnetworks, associated with multiple tasks, exhibit partial overlap and undergo parallel training. (Chen et al., 2021b) investigate whether subnetworks extracted from pre-trained computer vision models maintain downstream transferability. Their experiments verify that in pre-trained models for various tasks, matching subnetworks of different sparsity levels can consistently be identified. These subnetworks demonstrate universal transferability to downstream tasks without performance degradation. However, even within the same domain of studying LTH in computer vision (CV), (Girish et al., 2021) indicate that the acquired tickets from ImageNet pre-training do not transfer effectively to downstream tasks, particularly in the context of object recognition, instance segmentation, and keypoint estimation.

(Iofinova et al., 2022) investigate the transferability of winning tickets in the context of pruned convolutional neural networks (CNNs) trained on ImageNet. They consider various pruning methods and find that sparse models can match or surpass the performance of dense models in transfer tasks, even at high sparsity levels. (Liu et al., 2022b) discover that the preserved pre-training performance within the subnetworks is the reason behind the success of magnitude pruning, which is linked to downstream transferability. As a result, they propose a novel approach to adjust the model architecture during the pre-training phase, aiming to retain more transferability. By training binary masks over model weights on the pre-training tasks, the method aims to preserve the universal transferability of the subnetwork across various downstream tasks. Compared to conventional magnitude-based pruning, this technique significantly improves the overall performance of identified BERT subnetworks on downstream tasks. Additionally, it proves to be more efficient in subnetwork identification and particularly advantageous for fine-tuning with limited data availability.

In addition, meta-learning, which focuses on training models to quickly adapt to new tasks with minimal data, is closely related to transferability. (Gao et al., 2022) explore the application of the lottery ticket hypothesis (LTH) in the context of meta-learning to achieve efficient learning with reduced computational cost. They demonstrate that there exist sparse sub-networks, referred to as "meta winning tickets," which can be trained for few-shot classification tasks with performance comparable to the original network. The utilization of LTH in meta-learning facilitates the adaptation of networks on resource-constrained IoT devices. The paper also introduces a novel approach for the early detection of meta-winning tickets, enabling efficient training on devices with limited resources.

(Burkholz et al., 2022) further substantiates that winning tickets can be applied across various tasks, suggesting a level of universality. This notion is formally established and theoretically validated, demonstrating the existence of universal tickets that do not necessitate additional training. Leveraging these validation results,

they introduce several novel pruning approaches for strong lottery tickets and explicitly outline the sparse structure of the universal function family. Subsequently, (Huang et al., 2022) apply the LTH to CNNs integrating plug-and-play self-attention modules (SAMs). Their investigation reveals that complete integration of SAMs with all blocks may not lead to maximal performance enhancement. Consequently, they introduce the concept of Lottery Ticket Hypothesis for Self-attention Networks, suggesting that a self-attention network encompasses a sparse subnetwork capable of expediting inference, reducing parameter overhead, and maintaining accuracy. This conjecture is substantiated by a combination of empirical observations and theoretical underpinnings.

(Chen et al., 2021f) attempt to tackle a challenge: transferring winning tickets from one architecture to another network with a different structure. This endeavor aims to directly provide the latter with a winning ticket, thus eliminating the necessity for expensive iterative training. Therefore, they propose the Elastic Lottery Ticket Hypothesis (E-LTH). It suggests that through strategic replication, removal, and re-ordering of layers within a network, the corresponding winning ticket can be extended or compressed to fit into a deeper or shallower network from the same family. This subnetwork's performance closely rivals that of the latter network's winning ticket directly identified by IMP.

### 6.4 Dual Lottery Ticket Hypothesis

Dual Lottery Ticket Hypothesis (DLTH) (Bai et al., 2022) posits that a randomly pruned model can be transformed into a winning ticket that achieves comparable performance as the original model through regular fine-tuning. The hypothesis indicates that any subnetwork can be a winning ticket with careful fine-tuning. DLTH employs the Random Sparse Network Transformation (RST) method to obtain sparse subnetworks. This process bypasses the need for pre-training the model and directly selects a random subnetwork from the randomly initialized network. Subsequently, it utilizes a regularization-based method to extrude information from other weights that are masked to target the sparse structure. Consequently, the training process of DLTH is more efficient than that of LTH.

In the process of information extraction, a loss function similar to that in Continuous Sparsification is employed, but with a switch from $L_1$ regularization to $L_2$ regularization:

$$L_R = L(f(x; \theta), D) + \frac{1}{2}\lambda\|\theta^*\|_2^2, \tag{4}$$

where $L_R$ is composed of the regular training loss $L$ and an $L_2$ regularization term, $\theta^*$ denotes the masked weights.

Building upon this, the authors continually increase the trade-off parameter $\lambda$ during the optimization of $L_R$:

$$\lambda_{p+1} = \begin{cases} \lambda_p + \eta, & \lambda_p < \lambda_b, \\ \lambda_p, & \lambda_p = \lambda_b, \end{cases} \tag{5}$$

where $\lambda_p$ represents the regularization term at the $p$-th updating, $\eta$ is the given mini-step for gradually increasing $\lambda$, and $\lambda_b$ is the bound to limit the increase of $\lambda$. Through this process, the information of $\theta^*$ is gradually squeezed into the unmasked weights, denoted as $\bar{\theta^*}$. The importance of weights dynamically adjusts, transitioning from a balanced state (where all weights are equal) to an imbalanced state (as the magnitude decreases, $\theta^*$ becomes less significant). After enough extrusion, the influence of $\theta^*$ on the network becomes extremely limited, and they are subsequently removed to obtain the final sparse network.

Furthermore, LTH's sparsification relies on the mask obtained from the pre-trained network, resulting in a fixed sparse structure. In contrast, DLTH depends on the randomly selected network, offering greater flexibility in the selection of sparse subnetworks. When compared to Pruning at Initialization (PI), DLTH focuses on transforming randomly selected subnetworks into trainable states, rather than starting from scratch to select a specific subnetwork. Thus, the sparsity structure in DLTH is controllable. In terms of accuracy, ResNet56 pruned using the RST method outperforms ResNet56 pruned using LTH at sparsity ratios of 50%, 70%, 90%, 95%, and 98% on the Cifar-10 and Cifar-100 datasets.

In summary, DLTH introduces a novel approach for training sparse networks that surpass existing methods (LTH and PI) in terms of generality, efficiency, accuracy, and flexibility in sparse structure selection. Inspired by the Lottery Ticket Hypothesis and Dual Lottery Ticket Hypothesis DLTH (Bai et al., 2022), (Wang et al., 2022a) propose two novel algorithms, UniLTH and UniDLTH, aimed at unifying LTH and DLTH. UniLTH modifies the LTH approach by employing progressively increasing regularization on all weights of the neural network to discern their respective importance. On the other hand, UniDLTH applies to grow $L_2$ regularization exclusively to the masked weights for information extrusion. Both UniLTH and UniDLTH consist of two stages. In the first stage, the neural network is trained without any regularization, and early stopping is employed with a patience parameter, denoted as $\phi$. When the validation loss fails to decrease for $\phi$ consecutive epochs, training is halted, and parameters are reverted to their state from $x_i$ epochs earlier. In the second stage, iterative pruning is integrated with increased regularization to search or transform winning tickets. The network is alternatively trained with increased regularization and pruned until it attains the desired sparsity ratio.

Experimental results demonstrate that after applying UniLTH and UniDLTH to prune ResNet50 at various sparsity ratios, the performance surpasses that of LTH and DLTH under equivalent experimental conditions.

## 7 Relation with other topics

Extensive work has been done to fill the gap between LTH and other topics including robustness, scalability, and fairness. In this section, we summarize the gap and the experimental results.

### 7.1 Robustness

Although winning tickets can achieve remarkable performance, these subnetworks manifest vulnerability to adversarial attacks (Zheng et al., 2022). To tackle this issue, the authors in (Zheng et al., 2022) initially convert the binary mask into a hard concrete distribution. This renders the mask differentiable, allowing for continuous adjustments during training based on the objective. Furthermore, it incorporates an adversarial loss objective to guide the search for robust tickets, prioritizing both accuracy and robustness. In contrast to adversarial training, (Liu et al., 2022a) propose a novel approach involving randomly initialized binary networks with fixed parameters (+1 or -1) to identify a subnetwork structure resilient to attacks. By adaptively pruning various network layers, employing an effective binary initialization strategy, and integrating a final batch normalization layer for enhanced training stability, they achieve more compact networks with competitive performance compared to existing methods.

The limitations of adversarial learning are also highlighted in (Xi et al., 2022). This method requires the generation of adversarial samples through gradient descent, making it more computationally expensive compared to conventional fine-tuning. By investigating the optimization process of adversarial training, it was discovered that resilient connectivity patterns emerge early in training well before parameters stabilize. Therefore, the authors devise a two-step training approach to obtain robust early-bird tickets: (1) early-stage search for robust tickets with structured sparsity; (2) subsequent fine-tuning of these robust tickets. (Chen et al., 2022b) introduce a more stringent concept known as "Double-Win Lottery Tickets." They employ the approach of robust pre-training to extract subnetworks from the network. These subnetworks can independently transfer to various downstream tasks while achieving both standard and robust generalization, mirroring the full pre-trained model's capabilities.

### 7.2 Fairness

There is also a gap between fairness and LTH. Hansen & Søgaard (2021) argue that the use of lottery tickets may impact the fairness of machine learning models. They assess the fairness of extracting lottery tickets through layer-wise and global weight pruning in the field of natural language processing. Experimental results indicate that at high pruning rates, there is a slight increase in group disparities. Models trained with a distributed robust optimization objective are sometimes less sensitive to pruning, but the results are inconsistent. To alleviate unfairness with sparse networks, (Tang et al., 2023) propose to find appropriate binary masks for the weights to obtain fair sparse subnetworks. It turns out that such sparse subnetworks

with inborn fairness exist in randomly initialized networks, achieving an accuracy-fairness trade-off. They theoretically provide fairness and accuracy guarantees for fair scratch tickets and empirically verify the existence of fair scratch tickets on various datasets.

### 7.3 Federated Learning

Federated learning (FL) is a collaborative decentralized learning paradigm to address privacy in machine learning. It suffers from heavy communication such as uploading and downloading large volumes of parameters in the model. LTH is introduced to reduce both the communication and computation costs through model compression (Itahara et al., 2020). The key idea is to obtain a winning ticket from the original model. LotteryFL (Li et al., 2020) further assign each client with a personalized winning ticket. Following LotteryFL, CELL (Seo et al., 2021) exploits downlink broadcast for communication efficiency and introduces a novel user grouping method to mitigate stragglers. FedLTN (Mugunthan et al., 2022) proposes post-pruning without rewinding to achieve faster and greater model sparsity in FL. Babakniya et al. (2023) identify a challenge in federated learning (FL) on resource-limited edge nodes due to constrained computation and communication capabilities. They explore the use of off-the-shelf sparse learning algorithms, which train a binary sparse mask on each client, aiming to achieve a consistent sparse server mask with sparse weight tensors. However, their investigation reveals that this approach leads to a significant drop in accuracy compared to FL with dense models, particularly for clients with limited resources. They observe a notable lack of consensus among the trained sparsity masks on clients, hindering convergence for the server mask and potentially causing a substantial performance decline. In response, the authors propose "federated lottery aware sparsity hunting" (FLASH), a unified sparse learning framework designed to enable the server to attain a sparse sub-model, maintaining classification performance under highly resource-limited client settings. Additionally, they introduce "hetero-FLASH" to accommodate FL on different devices with varying parameter density requirements based on their resource constraints.

### 7.4 Reinforcement Learning

Due to the distributional shift inherent in reinforcement learning (RL) problems, the performance of winning lottery tickets can be affected (Vischer et al., 2022). Specifically, the feed-forward networks trained with behavioral cloning compared to reinforcement learning can be pruned to higher levels of sparsity, suggesting that the RL agents require more degrees of freedom. The analysis shows that the effect in learning paradigms can be attributed to the identified mask rather than the weight initialization. Also, the input layer mask selectively prunes entire input dimensions that turn out to be irrelevant to the task. The existence of winning tickets has also been investigated in a number of discrete-action space tasks, including both classic control and pixel control (Yu et al., 2019). The result confirms that winning ticket initialization outperforms parameter-matched random initializations at extreme pruning rates in RL.

## 8 Applications

### 8.1 CV and NLP

VGG and ResNet are the two most widely used models to investigate LTH in the literature. The existence of winning tickets is verified on two benchmark datasets: MNIST and CIFAR10. Besides these classification models in computer vision, (Girish et al., 2021) apply LTH to object recognition tasks. The authors empirically investigated the applicability of LTH for model pruning in tasks related to object detection, instance segmentation, and keypoint estimation. Additionally, the research delves into the behavior of trained tickets in relation to attributes like object size, frequency, and detection difficulty.

Large-scale pre-trained language models such as transformers are usually over-parametrized (Liang et al., 2021a). LTH has been applied to the GLUE benchmark, a collection of tools for evaluating the performance of models across a diverse set of existing natural language understanding tasks. These tasks include question-answering, sentiment analysis, and textual entailment, and an associated online platform for model evaluation, comparison, and analysis.

Recently, multimodal learning has been revolutionized by combining computer vision and natural language processing technologies. An empirical study has been performed to assess whether winning tickets also exist in pre-trained vision-and-language (VL) models Gan et al. (2022). The authors consolidate 7 representative VL tasks: visual question answering, visual commonsense reasoning, visual entailment, referring expression comprehension, image-text retrieval, GQA, and NLVR. It turns out that the sparse model can not strictly match the performance of the full model. However, a "relaxed" winning ticket at 50%-70% sparsity can reach 99% of the full accuracy.

### 8.2 Deployment

Obtaining winning tickets for large models involves a significant computational cost. Consequently, these winning tickets are regarded as valuable assets, warranting the need to safeguard their copyright. (Chen et al., 2021g) present a distinctive approach for validating lottery tickets by leveraging sparse topological information. This is achieved through the development of various graph-based signatures that can be embedded as credentials. By integrating trigger-set-based methods, the proposal can be applied in both white-box and black-box validation scenarios. MSSR (Lin et al., 2023) introduces a single adaptable model that encompasses multiple SR models of varying sizes. The MSSR framework involves two stages: a forward stage for model compression and a backward stage for expansion. In the forward stage, LTH with rewinding weights is used to gradually reduce the size of the SR model, employing pruning-out masks that form nested sets. Additionally, stochastic self-distillation (SSD) is utilized to enhance the performance of sub-networks. In the backward stage, the smaller SR model is expanded by reinstating and fine-tuning the pruned parameters based on the pruning-out masks acquired in the forward stage.

## 9 Open Issues

### 9.1 Acceleration in Practice

The irregular sparse patterns in a winning ticket pose a challenge to accelerating on hardware because most of the accelerators are optimized for dense matrix operations. It will limit the practical use of unstructured pruned subnetworks even if the sparsity level is very high. Therefore, it is urgent to design effective algorithms that can find structured winning tickets or transform an unstructured subnetwork into a structured subnetwork without compromising performance. Another solution is to design hardware accelerators that can support sparse matrix operations. We remark that existing works use sparsity or FLOPs (floating point operations per second). A new performance metric is needed to evaluate the real running time.

### 9.2 Theoretical Understanding and Design Better Network.

The identified winning tickets have the potential to improve our theoretical understanding of neural networks and help us design better networks. The original, overparameterized model has too much complexity and may result in overfitting. The lottery ticket hypothesis offers a complementary perspective on the relationship between compression and generalization: a larger network might explicitly contain simpler representations. By pruning, we can find an approximated simple (e.g., linear) model. However, the potential implications for theoretical study of optimization and generalization have not been studied yet. Also, it is unclear how to design a better model based on the initialization of winning tickets and sub-structure.

### 9.3 LTH in Diffusion Model.

Although diffusion models have shown impressive performance in capturing distributions and sample quality, they are notoriously slow to generate data due to the long chain of reversing the diffusion process. It is intuitive to mitigate the computational cost by pruning the reverse model. An open question remains: Can we use sub-networks with different sparsity in the reverse process? Since a winning ticket can be considered an equivalent version of the original model, we can leverage a sparser sub-network in the late stage of denoising to further improve efficiency. Intuitively, the noise in the later reverse process has been reduced and it will be easier to optimize. The challenge lies in how to optimize a dense model and a winning ticket while

guaranteeing the convergence of training. Another challenge is to decide at which step to use the winning ticket. We leave this open question for future investigation which we hope will stimulate the research on improving the efficiency of the diffusion model.

## 10 Conclusion

This survey for the first time investigates the lottery ticket hypothesis in the literature. In this paper, we first present a taxonomy of recent research that categorizes existing works into 8 topics: theory, special models, experimental insight, algorithms, efficiency, relations with other topics, open issues of existing works, and applications. For each topic, we conclude the contribution of the existing works and analyze the potential issues. In addition, we identify the publications with available codes online. Lastly, we discuss the limitations of LTH and point out a few potential research directions. We expect this survey can assist future researchers in seeking pruning technologies for various models and applying LTH in real applications.

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
