# OpenReview forum: "A Survey of Lottery Ticket Hypothesis"
_TMLR — Rejected by TMLR_

### Review · Reviewer_tF79 · 2024-06-19

**Summary Of Contributions:**

As clearly stated upfront, the manuscript surveys the research on the lottery ticket hypothesis. To organize the survey, it is organized into eight different topic areas.

**Audience:**

Yes

**Broader Impact Concerns:**

None.

**Claims And Evidence:**

Yes

**Requested Changes:**

- The LTH hypothesis is mainly interesting because the phenomenon seems to tell us something about how a neural network (NN) learns during training. For practitioners who want to build effecient NN models, there is a whole literature on sparsity, of which LTH is just a subset.  This is never addressed in the manuscript. I believe it should be.
- For example, are the methods proposed for finding sparse subnetworks useful within the larger context of building sparse, resource efficient NNs? Or are these methods only for the sake of our theoretical understanding?
- The abstract describes the LTH as that a DNN contains a subnetwork "that can achieve even better performance than the original model". This contradicts the definition given on page 3, which says "can reach similar accuracy".
- There are various grammatical problems throughout. In Section 5.3: "pruning is known to have an inability for real hardware acceleration." Section 7: "we conclude these connections."  The spelling "re-training" is used.
- Descriptions are unclear. In Section 1: "Unstructured pruning results in sparse individual parameters without pre-defining the positions of zeros." In Section 6.3: "These tickets can be directly reused across various tasks without the need for retraining." What kind of retraining? I couldn't understand from the context. In Section 6.2: "This subset accurately mirrors the performance disparities observed under various training frameworks, hyperparameters, and other conditions." Also, the description of sparsifying image patches at the end of Section 6.2 requires more explanation. Description of the dual lottery ticket hypothesis in Section 6.4 was unclear.
-  I thought the authors' could have used their expertise in this field to help synthesize some of these results.

**Strengths And Weaknesses:**

Strengths:
- The manuscript delivers on what the title promises. As far as I know, this kind of summary does not exist, and it will likely be useful to anyone looking for a survey of papers in the field.
- The proposed topic areas mostly make sense for organizing the existing literature.

Weaknesses:
- This is not a good introduction to the LTH for researchers outside the field (e.g. graduate students).
- The manuscript fails to synthesize any new understanding.
- Descriptions of the papers is shallow and uncritical.
- Some papers are summarized to the point that it is difficult or impossible to understand what the paper is about.
- There are some grammatical errors.
- The relation to research on building sparse neural networks is never explained.

---

### Review · Reviewer_11Mx · 2024-10-06

**Summary Of Contributions:**

The paper offers an overview of the most important references related to the topic of the "Lottery Ticket Hypothesis".
It categorizes the existing literature in a number of categories, and mentions the contributions of the several cited works in a compact way probably accessible to experts.

**Audience:**

Yes

**Broader Impact Concerns:**

No concerns

**Claims And Evidence:**

No

**Requested Changes:**

The authors should choose their goal.
* If they intend to make a good survey paper, they should explain all terms used, ideally in a way accessible to non-experts interested in getting an understanding of the Lottery Ticket Hypothesis domain.
* If they intend to only provide a set of references to the most important papers as a useful tool to experts who are already familiar with this field, they should adapt the title and the claims may, e.g., "Overview of key references about the Lottery Ticket Hypothesis" would be a fitting title.  If a reviewer who is expert in this line of work can confirm the adequacy of the overview of existing work, this may be a viable strategy where it is clear for potential readers what to expect.

In its present form, the paper makes claims (providing a full survey) which are not supported by the content of the work.

**Strengths And Weaknesses:**

The Lottery Ticket Hypothesis is an interesting and emerging topic, so a survey paper would be appropriate and timely.

I'm not an expert in this specific line of work and can't assess in a reasonable amount of time the completeness of the overview nor the correctness of what is said about existing literature.

Minor: Some text gives the impression that the paper is not very rigorous and the structure can be improved, but I can't say with much confidence.  E.g., Sec 2 starts saying in $f(x,\theta)$ the $x$ are "training samples" while hopefully the function also can be applied to test samples.  Then a three-step procedure (pretraining, pruning, re-initialization) is presented, after which a paragraph follows which seems to suggest also several alternative approaches.

The paper does not provide an introduction for non-experts.  Contributions of cited works are summarized, using terminology which is not defined in the present paper.  As such, if the overview would be complete and correct then it may be a nice reference for experts on this specific topic.

A good survey paper provides an introduction on the relevant concepts to the reader, and discusses the advantages and disadvantages of the several lines of thinking which exist.  This is not what the current paper does.  For example
* The text refers to Iterative Magnitude-based Pruning but does not give a hint what it is.
* It is unclear what is a target subnetwork
* The text refers to straight-through estimator without introducing it
* The text suggests some papers try to prove some bounds, but it is unclear why kind of bounds they try to prove

---

### Review · Reviewer_AAnb · 2024-11-03

**Summary Of Contributions:**

This survey provides a comprehensive introduction of lottery ticket hypothesis, which refers to the existence of a subnetwork of much smaller size in a large network with comparable (or even better) performance. The paper defines clearly on the lottery ticket hypothesis and introduces different lines of work around it, including theory, relevance to network architectures, algorithms for finding the winning ticket, and connection to broader concepts.

**Audience:**

Yes

**Broader Impact Concerns:**

No concern.

**Claims And Evidence:**

No

**Requested Changes:**

It is really nice to categorize the discussion on lottery ticket hypothesis into 8 topics on Page 2. I am wondering what are the inner connections between these topics and are there particular reasons to consider these 8 topics instead of potentially others?

I found it hard to parse the information provided after "The mechanism behind finding winning tickets using iterative magnitude-based pruning (IMP) is identified as three key inquiries" on Page 3. This sentence seems to be a bit disconnected with the context around it. Moreover, the explanation following this sentence consists of many jargons for me to understand.

Turning towards the introduction of theoretical study on Page 3, the sentence "By approximating ReLU networks, This work proves that optimizing the weights is equivalent to pruning entire neurons of a sufficiently large random network" seems unclear to me. What is approximating ReLU networks referring to and how does it connects to optimizing the weights? In addition, does the result depend on the optimization algorithm and the loss function?

Following the above, in "a sufficiently over-parameterized neural network with random weights contains a subnetwork with roughly the same accuracy as the target network, without any further training. However, the polynomial over-parameterization ...", polynomial overparameterization might need a definition.

In "An exponential improvement to the over-parameterization requirement is offered in Pensia et al. (2020)" on Page 3, what does exponential improvement mean here? The previous sentence only says polynomial overparameterization.

The sentence "Instead of providing significantly tighter bounds, (Orseau et al., 2020) points out that the overparameterized network only needs a logarithmic factor number of neurons per weight of the target subnetwork" is not very clear. What is the logarithmic factor? "per weight" should be "per layer"?

The sentence "This discovery implies effective lottery tickets can be identified at frequently used depths, necessitating only logarithmic levels of overparameterization" needs further elaboration. From my understanding, lottery ticket hypothesis says that a smaller subnetwork can be extracted from a pre-trained larger network. However, the existing theory shows the existence of a smaller network inside a larger network, which yields approximately the same level of accurate outputs when the weight parameters are randomly initialized. If I were correct, there is still a gap between theory and the lottery ticket hypothesis.

I am not following the sentence "The strong lottery ticket hypothesis is extended functions that maintain the operation of the group, known as G-equivariant networks". What do you mean by extended functions?

The likelihood in this sentence "The study establishes, with a high likelihood" is confusing. Do you want to say "with a high probability"?

**Strengths And Weaknesses:**

I believe the authors have made great efforts in summarizing different aspects of lottery ticket hypothesis. The paper is well organized and easy-to-follow for audiences.

Table 1 can be particularly helpful for researchers or incoming researchers in the field, which lists information of existing works with code.

I have some comments around the introduction and theory sections, which is detailed in the requested changes section. Due to my background, I am not able to provide a comprehensive review on all the aspects of lottery ticket hypothesis. I believe other reviewers will provide valuable comments.

---

### Decision · Action_Editor_rrvZ · 2025-03-14

**Recommendation:** Reject

**Comment:**

While the paper does a reasonable job providing an overview of existing results relating to the Lottery Ticket Hypothesis, the reviewers express doubts--and I tend to agree--regarding whether the paper would be accessible to researchers who are not experts in the field. This is not a desirable property for a survey paper, and some reviewers are very clear that this should not be certified as a survey. Given that and the lack of original content, the recommendation is that the paper be rejected.

**Audience:**

Yes, the paper will be of interest to researchers who are interested in the lottery ticket hypothesis, but even more generally to researchers who are interested in sparsity, improved inference performance, compression, etc. when relevant to large neural network models.

**Claims And Evidence:**

This is primarily a survey paper, so to the extent that there are claims in the paper, these are claims that already exist in the literature. Citations to the literature are reasonably comprehensive.